# Existing Evidence for Influenza B Virus Adaptations to Drive Replication in Humans as the Primary Host

**DOI:** 10.3390/v15102032

**Published:** 2023-09-30

**Authors:** Matthew J. Pekarek, Eric A. Weaver

**Affiliations:** Nebraska Center for Virology, School of Biological Sciences, University of Nebraska-Lincoln, Lincoln, NE 68583, USA; mpekarek2@huskers.unl.edu

**Keywords:** influenza B virus, host adaptation, virus–receptor interaction, nuclear replication, immune evasion, viral evolution

## Abstract

Influenza B virus (IBV) is one of the two major types of influenza viruses that circulate each year. Unlike influenza A viruses, IBV does not harbor pandemic potential due to its lack of historical circulation in non-human hosts. Many studies and reviews have highlighted important factors for host determination of influenza A viruses. However, much less is known about the factors driving IBV replication in humans. We hypothesize that similar factors influence the host restriction of IBV. Here, we compile and review the current understanding of host factors crucial for the various stages of the IBV viral replication cycle. While we discovered the research in this area of IBV is limited, we review known host factors that may indicate possible host restriction of IBV to humans. These factors include the IBV hemagglutinin (HA) protein, host nuclear factors, and viral immune evasion proteins. Our review frames the current understanding of IBV adaptations to replication in humans. However, this review is limited by the amount of research previously completed on IBV host determinants and would benefit from additional future research in this area.

## 1. Introduction

Severe influenza infections in humans are typically caused by two influenza virus types: influenza A viruses (IAV) and influenza B viruses (IBV). These infections are suspected to cause up to 5 million severe cases of influenza disease annually [1]. In the 1980s, two antigenically distinct lineages of circulating IBV were identified using reference antiserum neutralization assays [2]. More recently, it was proposed that amino acid changes near the receptor binding site are responsible for the divergence of the two IBV lineages [3]. The lineages are characterized based on their relationship to two reference strains, B/Victoria-like (B/Vic) and B/Yamagata-like (B/Yam). Co-circulation of both lineages necessitated the development of quadrivalent vaccines to better protect against circulating IBV [4,5,6]. Severe disease burden in vulnerable populations has driven a public health pursuit to improve protection against IBV [7,8,9].

Throughout recorded history, IAV has been responsible for four major pandemic outbreaks, in 1918, 1957, 1968, and 2009. These pandemics were caused by the H1N1, H2N2, H3N2, and H1N1 subtypes, respectively [10]. The strains responsible for these pandemics had not previously circulated in humans, resulting in a lack of pre-existing immunity in human populations [11]. The lack of pre-existing immunity subsequently allowed nearly unlimited access to susceptible human hosts and unharnessed human-to-human transmission. Despite sharing many similar proteins to IAV, shown in Figure 1, IBV has not been responsible for pandemic influenza outbreaks. Since its discovery, IBV has been primarily isolated from human hosts. This is a stark contrast to IAV, whose host range consists of birds, swine, and other mammals [12]. Infectious IBV has been isolated from harbor seals, but phylogenetic similarity to human strains circulating at the same location and time suggests the seals were infected with human IBV strains [13]. Therefore, it is likely that humans remain the sole reservoir driving IBV circulation. Consistent IBV circulation in humans likely maintains a level of pre-existing immunity in the population and may contribute to the lack of an IBV-caused pandemic.

Due to shared structural and replicative properties with IAV, we hypothesized that the distinct difference in host range between IBV and IAV is unlikely to be due to chance alone. Viral and host factors important to IAV host determination have been reviewed previously [12,14]. However, to the best of our knowledge, no reports have reviewed possible IBV adaptations to human hosts. Because of this, we chose to review the relevant findings suggesting IBV circulation in humans is due to a restriction of host range rather than by chance. We focused on three main areas most likely to contain host restriction factors due to their crucial role in the viral replication cycle: hemagglutinin-receptor interaction, nuclear replication factors, and immune evasion mechanisms. While some evidence to support this hypothesis has been reported across various stages of the viral replication cycle, including receptor binding, we found that research into this area was more limited than we expected. Therefore, this review focuses on the factors that could suggest IBV host range restriction to humans. We discuss how factors identified as host determinants of other influenza viruses could be investigated for a potential role in IBV host restriction. By conducting this literature review, we hope to provide the field of IBV research a framed view of various factors that may restrict IBV replication to humans and prevent circulation in other common hosts of influenza viruses.

## 2. Hemagglutinin-Specific Adaptations

In the initial stages of infection, IBV must bind to sialic acid residues on the surface of the cells lining the respiratory tract. Receptor binding is facilitated by the IBV hemagglutinin protein (B-HA). This receptor-binding interaction is crucial for successful viral infection and plays a major role in host specificity of IAV [15,16,17]. The receptor binding site of the B-HA is formed through contact of the 140-loop, 160-loop, 190-helix, and 240-loop. The tertiary receptor binding site then interacts with the sialic acid linkages on the surface of respiratory cells [18]. Human influenza viruses, including both lineages of IBV, bind preferentially to α2,6 linked sialic acid residues [19,20,21,22,23,24]. Avian IAV strains preferentially bind α2,3 linked-sialic acids [12,25]. The high abundance of α2,6 linked sialic acid residues in the human upper respiratory tract [16,26,27] provides ample access to susceptible host cells during viral infection. Among IBV, only the B/Victoria-like lineage has been linked with binding α2,3 linked sialic acid residues [20]. Interestingly, it was recently proposed that younger individuals have an increased abundance of α2,3 linked sialic acid residues in the lower respiratory tract compared to adults [28]. This difference could be associated with reports of more common and severe Victoria-like IBV infections in younger individuals than Yamagata-like infections [7,9,29,30]. Currently it is unknown whether the amino acid mutations responsible for the lineage divergence play a role in the difference of sialic acid receptor binding. The bias of B-HA binding α2,6 linked sialic acids may block transmission into hosts whose upper respiratory tracts contain more α2,3 linked sialic acids.

To facilitate membrane fusion between the viral envelope and the host endosome after receptor interaction, B-HA must have been enzymatically activated [31]. While other viruses commonly encode their own proteases [32], influenza viruses rely on host proteases to activate their HA proteins. The transmembrane serine protease 2 (TMPRSS2) is a host protease responsible for enzymatic activation of influenza viruses [33,34,35], paramyxoviruses [36], rotaviruses [37], and coronaviruses [38]. IAV is thought to be highly reliant on TMPRSS2 for efficient viral infection [33,39]. This was also observed when low pathogenic avian IAV strains were used to infect mouse and human airway epithelial cells [40]. Activation by TMPRSS2 and other trypsin-like proteases highly represented in the airways of humans and the gastrointestinal tract of waterfowl, may support the wide host range of IAV [41]. In comparison to IAV, IBV can be activated by a much broader range of human proteases [34,35,42]. Many of these additional proteases, including the kallikrein-family (KLK) proteases [35], are found in abundance in the human respiratory epithelium [34]. This abundance of proteases at the site of infection may point to an adaptation towards human airway infection rather than non-human hosts.

Outside of receptor binding and enzymatic activation, Laporte, et al. has shown that the physical environment of the human upper respiratory tract plays a key role in facilitating IBV infection. Interestingly, the pH required for efficient B-HA activation and membrane fusion was lower for IBV strains than for avian IAV (5.4–5.6 for IBV compared to 5.7–5.8 for avian IAV) [34]. Additionally, the optimal growth temperature of representative IBV strains was shown to be 33 °C, and the viral replicative yield is significantly reduced when the temperature increases up to 39 °C [34]. This is notable because the higher pH levels and temperatures tested in this study mimic the environments of the swine respiratory tract and the avian gastrointestinal tract, the major non-human reservoirs of IAV. Indeed, the avian IAV strains tested had optimal replication conditions that mirrored the avian gastrointestinal tract. Interestingly, the optimal growth conditions of the human IAV strains tested fell in between the IBV and avian IAV optimal growth conditions [34]. Given that the human respiratory tract is at a lower temperature and produces lower endosomal pH, this optimal growth condition observed, suggests an adaptation of IBV to human respiratory infection. Further, IBV replication is likely hindered by environmental factors specific to common non-human influenza hosts.

The receptor abundance in the human respiratory tract, the presence of enzymes capable of B-HA activation, and the optimal physical environment for IBV infection suggest possible host restriction of IBV replication to humans over non-human hosts. A summary of these factors can be seen in Table 1. However, the current lack of IBV research limits the impact of these observations. Overall, there are surprisingly few publications identifying adaptations of B-HA to the human respiratory environment. The few studies characterizing sialic acid binding affinities for B-HA are dated and fail to analyze strains isolated after 2010. Therefore, these studies do not consider the impact of newly emergent clades from both the B/Victoria-like and B/Yamagata-like lineages [43]. Interestingly, the binding affinity of the newest identified B/Victoria-like clade (V1A.3) [44], which contains a 3-amino acid deletion near the receptor-binding site [45,46], has yet to be characterized. This remains a gap in our understanding of possible restriction to human circulation among more recent IBV strains. Similarly, literature investigating B-HA enzymatic activation is based on older reference strains that no longer adequately represent the diversity of currently circulating IBV strains. Finally, little work has pursued the ability of avian- or swine-origin respiratory proteases to activate B-HA. Further work investigating this relationship and pursuing more recently isolated strains would help contribute to our understanding of how B-HA may be acting as a host restriction factor.

## 3. Nuclear Replication Adaptations

Influenza viruses belong to a group of RNA viruses that replicate their genome in the nucleus rather than the cytoplasm [47,48] despite not needing DNA replication machinery. The virus encodes its own RNA-dependent RNA-polymerase (RdRp), which is comprised of three viral protein subunits: PB2, PB1, and PA. These subunits interact to form the IBV polymerase complex (B-Pol). B-Pol lacks the critical function of 5′ mRNA cap production. Neither IAV nor IBV polymerase complexes have the ability to synthesize 5′ caps on their own [48,49]. Addition of a 5′ mRNA cap is required for stable mRNA translation by the host cell and for cap-dependent transcription of a complementary RNA (cRNA) molecule from viral genomic RNA (vRNA) [48,50]. To reach the nucleus, the IBV nucleoprotein (B-NP) contains nuclear localization signals (NLS) to facilitate transport to the nucleus [51]. Once the gene segments reach the nucleus, the virus usurps host factors to cross the nuclear envelope. The host factor importin-α binds three overlapping NLS on the N-terminal tail of B-NP to facilitate nuclear transport [51,52]. Importin-α has been previously described as a species-determinant for IAV [12,53]. However, a study has not been published testing this for IBV replication. Because there has not been a clear connection linking nuclear transport efficiency and replicating virus titers, we can only hypothesize that this may play a role in host restriction of IBV. Further investigation into this mechanism is required for a more definitive connection to IBV host restriction.

Once in the nucleus, B-Pol transcribes the viral RNA to initiate the viral replication cycle. Human acidic nuclear phosphoproteins (ANP32) are crucial for influenza RdRp activity [54,55,56,57]. The most important forms of ANP32 proteins in humans for influenza replication are ANP32A, -B, and -E. These proteins have many functions [58], but one noteworthy function is histone association and interaction with enzymes responsible for epigenetic post-translational modifications. However, influenza viruses are able to hijack these host proteins from their normal function to promote viral polymerase activity [57]. ANP32 proteins stabilize the influenza polymerase complex in a replicase promoting conformation prior to initiation of new RNA production [55]. Similar to IAV, both ANP32A and ANP32B can be utilized for efficient B-Pol activity. However, ANP32E can also support some polymerase activity in IBV but not IAV. Conversely, avian ANP32 proteins are unable to support efficient IBV polymerase activity but support human IAV polymerase activity [54]. Another study investigated the ability of other mammalian ANP32A and -B proteins for their ability to support B-Pol activity. Interestingly, their results showed that swine ANP32 proteins support B-Pol activity to a similar level as human ANP32 proteins. ANP32A and -B from horses, dogs, seals, and bats were only able to support B-Pol activity to a low level [56]. This reliance on mammalian ANP32 proteins may hinder IBV replication in birds but may not be a strong host restriction factor to human circulation alone. Further studies comparing the ability of human and non-human ANP32 proteins to stabilize B-Pol and promote viral replication could further support their role as a host restriction factor for IBV.

Again, few studies describe the interactions of IBV with host nuclear factors for the replication process. As mentioned briefly, the interactions of IBV with human importin-α focus on binding affinities but do not investigate how non-human homologs impact viral replication. These studies also fail to fully establish a mechanism of nuclear transport across the nuclear envelope. To support importin-α acting as a host restriction factor, future studies should assess the ability of different species’ importin-α to support IBV replication. This would provide evidence regarding to what extent IBV can utilize other species’ importin-α for nuclear transport to replicate the viral genome and express the viral proteins. While the importance of human ANP32 proteins was shown more clearly for IBV replication than importin-α, only avian ANP32 proteins were used to compare more than the two primary forms of ANP32 proteins. A major limitation to the studies showing ANP32 proteins supporting B-Pol activity was the absence of a comparison between the ability of host ANP32 proteins to support viral replication. Only B-Pol activity was measured in a minigenome reporter assay [56]. Another limitation is only the B/Florida/4/2006 strain (Yamagata-lineage) was used for this basic analysis, as IBV was not the primary focus of the study. Based on these limitations, we can only conclude that human nuclear factors may serve as possible host restriction factors at this time. Further dedicated studies testing the impact of these and likely other nuclear factors on IBV replication are needed to further support this conclusion.

## 4. Immune Evasion

A critical relationship between a host and pathogen is immune evasion. Influenza viruses specifically encode proteins to aid in immune evasion and utilize proteins with other viral functions to assist in evasion. A major difference between IAV and IBV in immune evasion is the lack of alternative polymerase protein products encoded in the PB1 and PA gene segments. IAV PB1-F2 [59] and PA-X [60] have immunomodulatory functions that support the replication of IAV [61,62,63], but these proteins are absent in IBV. Host-dependent expression differences in these accessory proteins have been described in IAV [22,64,65], suggesting these alternative products may play a role in the expanded host range. Therefore, there are other mechanisms employed by IBV to avoid the human immune system in the absence of these alternative polymerase factors.

The interferon (IFN) and interferon-stimulated gene (ISG) signaling pathways are a primary defense for the innate immune system in humans. During influenza infection, host cells produce IFN to activate host ISGs and limit total gene expression [66]. This impedes viral protein translation and prevents the production of progeny virions [67]. IBV infection strongly induces IFN expression in both human lung epithelial cell lines [68,69,70] and primary innate immune cells [71,72]. The IBV NS1 (B-NS1) protein has been shown to interfere with human IFN signaling through multiple mechanisms [73,74]. B-NS1 counteracts IFN signaling by preventing activation of the upstream transcription factor, IRF-3 [75]. A major B-NS1 target downstream of IFN signaling is the host IFN-stimulated gene 15 (ISG15) [76]. ISG15 is a ubiquitin-like modification protein that targets host genes to promote antiviral activity, including many that are also activated in response to IFN signaling [77]. B-NS1 specifically binds to human and non-human primate ISG15 but not other mammalian ISG15s [78,79,80]. B-NS1 can further modulate ISG15 activity by sequestering ISG15 target proteins [76,81]. This targeting of human ISG15 activity through multiple mechanisms suggests an important relationship between human ISG15 activity and IBV replication.

In addition to avoiding innate immune detection within the cell, the virus must also evade any adaptive immunity to infected cells. This recognition can come from cytotoxic T lymphocytes (CTLs). One way the virus attempts to avoid CTL recognition is through the downregulation of MHC Class I surface expression in infected cells. After IBV infection, cell lines stably expressing different HLA-alleles showed significantly decreased MHC Class I surface expression compared to IAV infection [82]. Another CTL evasion mechanism of IBV is a loss of an immunodominant epitope within a nuclear export signal (NES) in the IBV M1 (B-M1) protein [83]. This immunodominant epitope is found in the NES of IAV and may represent an adaptive mutation to avoid immune targeting in a conserved functional region [83]. While MHC class I downregulation is not unique to IBV infection [84,85,86], the apparent importance of human CTL evasion for replication may signal an advantage for IBV that could impact its ability to replicate in other hosts. However, little is known about IBV infections in this area. Further testing, comparing the impact of CTL responses on IBV and IAV infection between different hosts could uncover more concrete mechanisms relating immune evasion and host restriction.

We hypothesized that a virus with a limited host range such as IBV will be effective at evading its primary host’s immunity but not other host immunities. This would be due to the absence of viral responses to immune pressures exerted by non-human hosts, limiting virus replication. We also hypothesize that a virus with multiple mechanisms of immune evasion is more likely to have an expanded host range. The literature characterizing the B-NS1 interaction with human ISG15 supports our first hypothesis. As described above, B-NS1 is only able to rescue virus replication in the presence of human ISG15, while ISG15s from non-human species are able to inhibit replication in the presence of B-NS1. This suggests that IBV could be unable to successfully evade an aspect of non-human host immunity that prevents viral replication and spread within that host. We expect this species-specific targeting of ISG15, combined with other immune evasion mechanisms, plays a key role in promoting IBV circulation in humans while also potentially restricting replication in other non-human hosts.

Other immune evasion mechanisms have been characterized for IBV. Protein kinase R (PKR) [87,88], human myxovirus resistance protein 1 (MxA), melanoma differentiation-associated gene 5 (MDA5), and retinoic acid-inducible gene I (RIG-I) [89] have been described as targets of IBV immune evasion. Unfortunately, little has been reported about the ability of IBV to interfere with homologs of these proteins or signaling pathways from other hosts. Without the conclusions of these studies, we cannot adequately conclude what impacts these targets may have on IBV host restriction. If these targets do play a key role in host restriction, we would expect to see a level of species-specificity similar to what has been described with IBV and human ISG15.

## 5. Summary and Future Research Directions

The lack of an established non-human host presents a major difference between influenza B viruses and influenza A viruses. With the exception of isolated mammalian detection [13], no major animal outbreaks of IBV have been recorded. Despite this, IBV continues to circulate in humans. Here, we compile possible known host restriction factors that optimize IBV circulation in humans. To our surprise, limited studies have pursued this topic. Based on what is known about IBV replication, we then chose to review possible adaptations that may contribute to the lack of a non-human host of IBV. We summarize these possible host-specific adaptations in Figure 2.

The most extensively interrogated, possible IBV host restriction factor is the B-HA protein. The interaction between viral protein and host receptor is a crucial determinant of viral entry into susceptible cells. Unsurprisingly, host receptor characteristics likely shape whether or not that host can be infected. These characteristics are summarized in Table 1. A binding preference of B-HA to host α2,6-glycan structures drives efficient receptor binding to cells of the human respiratory tract. Further, IBV can utilize a wide range of human enzymes highly abundant in the human respiratory tract to activate the B-HA protein. Additionally, the optimal physical environment for IBV replication also seems tailored to the human respiratory tract compared to non-human species. Taken together, these factors suggest IBV adaptation towards cells with receptor structures and environments found in humans rather than non-human hosts.

Possible host restriction factors outside of B-HA are not as well understood. To determine if a host restriction phenomenon has been observed during IBV genome replication, we considered host nuclear factors. The importin-α protein family has been identified as a critical factor for transporting the vRNPs from the cytoplasm into the nucleus [51,52]. However, current literature only highlights the interaction between IBV vRNPs and human importin-α. Currently, it is unclear whether other species’ importin-α family proteins interact with IBV vRNPs to support IBV replication. Additionally, evidence from IAV replication studies suggests that the interactions between host importin-α proteins and the viral polymerase complex may help determine species specificity [90,91,92]. Therefore, it is possible that these proteins may have a similar host restriction role for IBV as well. Another family of nuclear proteins, the ANP32 protein family, has also been shown to be crucial in supporting IBV replication [54,56,57]. Unlike the importin-α protein family, studies have shown that B-Pol activity is optimized with mammalian forms of theANP32 family proteins rather than avian species’ ANP32 [54,56]. These results suggest that this protein family may be acting as a host species limiting factor rather than a host restriction factor.

IBV has also evolved several mechanisms to escape human immune recognition. B-NS1 drives multiple mechanisms that obstruct the human IFN response [73,74]. Targeting of human-specific ISG15 by B-NS1 suggests an integral role between IBV replication and the human IFN response during infection. IBV also encodes mechanisms to avoid long-lasting memory CTL responses by downregulating host MHC Class I molecules on the surface of infected cells [82] and accumulating mutations in the B-M1 protein’s sequence to protect a conserved functional region from an immunodominant CTL epitope [83]. This effort to evade different aspects of the immune system obviously relates to its importance in promoting viral replication. However, more detailed investigations analyzing the ability of IBV to actively evade immune recognition in other hosts would improve our understanding of this as a possible host restriction factor.

A major theme to this research topic is the relative lack of studies investigating the possible IBV host restriction factors. Numerous studies have identified host factors or host–viral interactions critical for efficient IBV viral replication. However, these studies often fail to compare the human factors with non-human homologs, leading to a gap in our knowledge of why IBV only circulates in humans. Table 2 provides an outline of the host factors known to be crucial for IBV replication. The open questions associated with these factors may provide further insights into their role in IBV host restriction.

Another potential avenue to discovery of IBV host restriction factors is considering the restriction factors of other types of influenza viruses. Factors that limit IAV, especially avian IAV, have been comprehensively reviewed [12,14,93,94]. Importantly, while strains of IAV can infect many different species, the host restriction factors identified often limit circulating strains to a particular host. Since these factors limit IAV crossover from one species to another, they may also play a role in host restriction of IBV. In fact, the importin-α and ANP32 protein families are described as host restriction factors for IAV as well [12,93]. However, much more is known about the role of these protein families for IAV host restriction than IBV restriction. As described above, more questions remain to clearly identify how IBV interaction with these proteins may impact host restriction. One host restriction factor well described for IAV with little attention for a role in IBV host restriction is the neuraminidase (NA) protein [14,95,96]. Various subtypes of IAV NA (A-NA) have been described to accumulate deletions in the stalk region when jumping from one host species to another [97,98,99]. This repeated deletion likely represents a critical mutation to overcome a host restriction factor to facilitate transmission in a new host species. Investigating this or other possible IBV host restriction mechanisms mediated by the IBV NA (B-NA) protein is critical to possibly understanding why IBV only replicates in humans.

Two other influenza types have also been discovered and circulate in mammals. Influenza C viruses (ICV) are distantly related to IBV and primarily infect humans [100]. Influenza D viruses (IDV) are closely related to ICV and were characterized as a second ICV subtype [101]. Both virus types differ from IBV and IAV genetically by encoding a hemagglutinin-esterase-fusion (HEF) protein [102] rather than distinct HA and NA proteins. ICV and IDV also bind to a chemically different receptor than IBV or IAV [103,104,105]. Wider natural host ranges for ICV and IDV compared to IBV have been described [106]. Unfortunately, little is known mechanistically about the host restriction factors for ICV or IDV. However, TMPRSS2 activates the ICV HEF protein similar to A-HA activation [107]. Host ANP32A protein has also been connected with ICV RdRp stability [55]. Additionally, ICV NS1 (C-NS1) and IDV NS1 (D-NS1) have been shown to possess IFN antagonistic properties [73,108]. The similarities observed between IBV, ICV, and IDV suggest similar factors may be contributing to host restriction and warrant further investigation. Uncovering these host restriction factors for one virus type may accelerate the discovery of the restriction factors for the others as well.

Despite an established host restriction environment for influenza viruses, host spillover events are detected. The most well-known examples are the pandemic spillover events from past influenza pandemics [10]. However, potential for reverse zoonoses, infections of animals from human circulation, is also possible. One reverse zoonosis pathway that is currently receiving attention is between humans and swine, most often in the H1N1pdm09 subtype [109,110,111]. The risk for reassortment and adaptation of these viruses to pigs remains a concern [112]. The SARS-CoV-2 pandemic has also opened a renewed interest in the impact of reverse zoonoses on global health [113,114]. While a spillover of IBV leading to sustained circulation in swine or other non-human hosts is unlikely, a clearer understanding of IBV host restriction could contribute to understanding barriers that prevent influenza and other viruses from undergoing frequent reverse zoonotic events.

A potential piece of non-molecular evidence supporting strong host restriction factors for IBV circulation is the possible extinction of the Yamagata-like lineage. The 2019–2020 flu season was predominantly characterized by the emergence of the new Victoria-like clade V1A.3 [44,46]. This led to low detection levels of Yamagata-like strains in the US early in the season [44], followed by an impact on influenza circulation due to non-pharmaceutical measures adopted to combat the spread of SARS-CoV-2 [115,116]. Despite influenza circulation rebounding in the more recent seasons [117], no Yamagata lineage detection since 2020 has led the field to consider whether the Yamagata lineage has possibly become extinct [118,119]. For a virus with a non-human reservoir, surveillance of the non-human hosts would likely detect virus even if human circulation was disrupted. However, for a virus that only circulates in humans, a break in human circulation would end the replication cycle and render the virus extinct. While we currently cannot rule out undetected circulation in an isolated population, the continued lack of detection of the Yamagata lineage suggests that the lineage may have become extinct. This lack of detection from humans and non-human hosts further supports the idea of only a single, restricted host population for IBV.

Based on our review of the available literature, there appears to be some evidence of IBV adaptation to human circulation throughout different stages of the replication cycle. Researchers have identified crucial interactions during receptor binding, genome replication, and immune evasion between viral and human factors that permit optimal virus replication to occur. The fact that these interactions occur throughout the virus replication cycle and not just during viral entry suggests that there is likely an important relationship between the human host and IBV replication. Many of these interactions require more comparison studies to investigate the impact that the factors from different host species have on IBV replication. This would help support the role of these factors as IBV host restriction factors. However, logically analyzing the available evidence suggests these adaptations could limit the ability of IBV to circulate in hon-human hosts. A better understanding of these interactions could unlock new information about how influenza viruses are restricted to their hosts. This information could then be leveraged as novel, specific antiviral targets in the future.

## Figures and Tables

**Figure 1 viruses-15-02032-f001:**
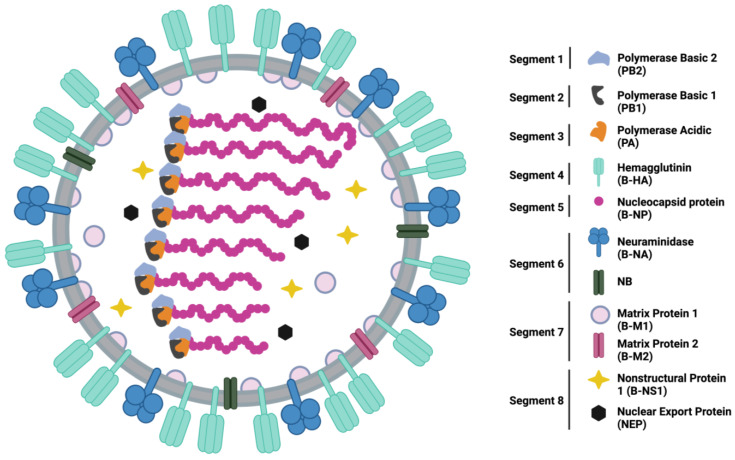
Schematic representation of the structure and organization of an IBV virion and gene segments with all 11 proteins represented. Schematic created in BioRender.

**Figure 2 viruses-15-02032-f002:**
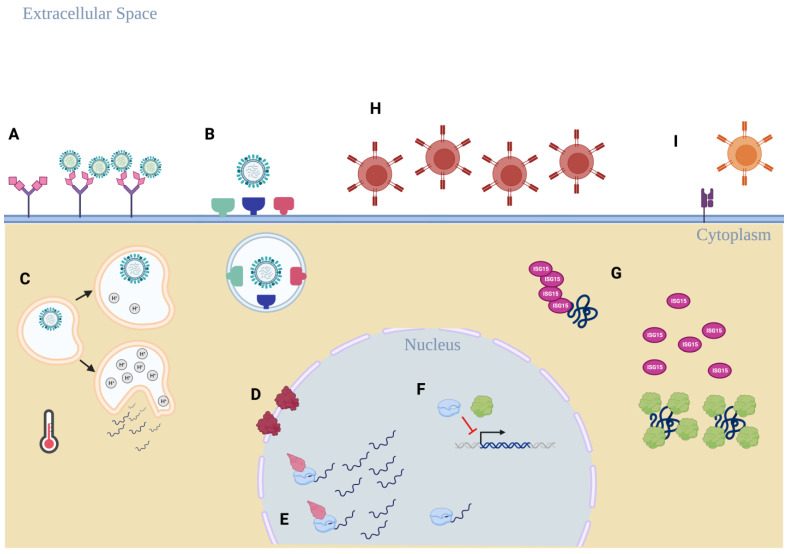
An overview of the possible host restriction factors that promote IBV circulation in humans. (**A**) B-HA binding preference for α2,6-linked sialic acid residues on glycoproteins more prevalent in the human respiratory tract. (**B**) IBV can utilize a broad range of host proteases found in abundance in the human respiratory tract and do not rely on TMPRSS2 for B-HA activation. (**A**–**C**) The physical environment in human respiratory epithelial cells and in the extracellular environment favors human replication, including human endosomal pH ranges and upper respiratory tract temperature. (**D**) IBV replication requires the human importin-α protein to be present to ensure functional replication takes place, but a role as a host restriction factor is currently unclear. (**E**) Human ANP32 proteins are required for optimal B-Pol efficiency, and a wider range of ANP32 proteins can support B-Pol activity compared to the IAV polymerase complex. (**F**) B-Pol subunit PB1 and B-NS1 disrupt the host IFN response by preventing expression of anti-viral transcription factors IRF-3 and STAT1, but this interaction has not been evaluated as a possible host restriction factor. (**G**) B-NS1 selectively prevents ISGylation of host proteins that promote anti-viral responses through sequestration of ISG15 target proteins. (**H**,**I**) IBV avoids T-cell recognition of infected cells through surface downregulation of host MHC class I presentation (**H**) and the lack of a broadly reactive CD8^+^ T-cell epitope found near the NES of B-M1 (**I**). Diagram created in BioRender.

**Table 1 viruses-15-02032-t001:** Summary of the known characteristics of B-HA and the human respiratory tract that point to possible host restriction to humans.

B-HA Role in Virus Infection	B-HA Function	Optimal Characteristic Found in Human Respiratory Tract	References
Receptor Binding	B-HA binds α2,6 linked sialic acid residues more efficiently than α2,3 linked sialic acid residues	Abundance and distribution of α2,6 linked sialic acid residues in the upper respiratory tract	[19,20,21,26,27]
Activation	B-HA must be enzymatically activated from its precursor form into its active subunits	A broad range of enzymes capable of activating B-HA are found in the upper respiratory tract	[34,35,42]
Membrane Fusion	B-HA undergoes pH-induced conformational change to fuse viral envelope and host endosomal membrane	Human endosomal pH and temperature of upper respiratory tract	[22,34]

**Table 2 viruses-15-02032-t002:** Summary of questions remaining to be answered about possible IBV host restriction factors.

Host Protein Family	Role in IBV Replication	Questions to Be Answered about Possible Host Restriction	References
Respiratory proteases	Activation of B-HA	Do homologs of human proteases known to activate B-HA similarly activate the glycoprotein?	[34,35,42]
Importin-α	Transport of vRNPs from cytoplasm into nucleus	Does the IBV vRNP interact with importin-α from non-human species? Can non-human importin-α proteins support efficient IBV replication?	[51,52]
ANP32	Support B-Pol complex activity	How well can non-human mammalian ANP32 proteins support IBV replication?	[54,56,57]
Innate immune sensors	Sensing of intracellular viral infection	Does B-NS1 interact with non-human homologs of innate immune sensing proteins? Can B-NS1 interaction lead to evasion of immune response in non-human hosts?	[73,74,75,87,88,89]

## Data Availability

No new data was created or statistically analyzed for this study.

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
