# Peer review of "Existing Evidence for Influenza B Virus Adaptations to Drive Replication in Humans as the Primary Host"

_viruses, 2023, doi:10.3390/v15102032_

Round 1
Reviewer 1 Report
Summary- this review attempts to summarise evidence for the host restriction of IBV. This is a unique perspective and it is agreed that more experimental evidence is required to understand this. However, revisions are required and overall the points made could be clearer.
Line 28-30- grammar issues
Line 31- antigenically characterized? Reference of the split since the 1980s?
Line 51-3- is it worth noting the 1977 reintroduction of H1N1, ‘psuedo-pandemic’?
Line 68- ‘accidentally’ this could be elaborated. What is implied?
Line 105- within or between? Within humans? Between species? Older humans can’t transmit to younger? The point is unclear
Line 132- B-HA compatibility and environment preference- rather than combined, doesn’t B-HA contribute to.
Can this subject of HA stability and physical environment be expanded. What is known for IAV- more stable Ph/ lower temperature is required for sustained transmission compared to avian IAV. Is this not common with IBV? What about differences in animal environments compared to humans- how would IBV fair in transmission. Is membrane fusion even the main driver of the pH stability? If so, why, are host restriction factors present?
Line 136- unusual or belong to a group of viruses that share this replication process?
Line 140- binds not ‘sit’?
Line 141- description is inaccurate or unclear- the ends of the GENOME rather than RNP, as the polymerase is part of the RNP, combined they make the RNP. The way it is described is unclear.
Line 142- repeats DNA replication machinery point.
Line 151- having given quite a lot of detail about mRNA transcription, why not reduce this detail and provide a short summary of replication also? What is the point trying to be achieved here? That IBV and IAV are similar? Perhaps better to remark if IAV has any host adaptations or not in these processes, compared to IBV?
Line 174- associated spelling. ANP32 proteins have many activities- ref https://onlinelibrary.wiley.com/doi/10.1002/bies.201400058
Line 185- The major mammalian:avian polymerase restriction as described here is very clear and the understanding is pretty good. However, there may be ADDITIONAL host factors to fine tune the polymerase… A point missed, is that IBV polymerase can function in avian cells. It is avian IAVs that cannot function in mammalian cells. So one may hypothesise that there is no major restriction in reverse zoonosis by the IBV polymerase.
Line 196- which protein?
Line 200- TLRs AND RNA helicases
Line 326- what evidence is there that IBV pol replicates preferentially with human ANP32s?
Figure 2- what about the extracellular environment- Fig 2C touches on HA stability- is this not also relevant to extracellular environment/ transmission?
This review reads as a general review of IBV in most places, as opposed to the intention of host restriction. Can this be made clearer throughout the review, with more direct conclusions or hypothesises provided, although this is better provided in the discussion. Perhaps some issue is the level of detail provided to make such a point of host restriction, whilst needed, can it be streamlined to better highlight the point being made?
Subjects not addressed by this review which should be considered:
- Can older Influenza B viruses cause pandemics from a lab escape, rather than non-human host?
- It seems remis to not comment on the possible extinction of the B-Yamagata lineage viruses since 2020. More evidence for lack of animal reservoir?
- What evidence of reverse zoonosis of IAV is there? Isn’t this also limited? There may be common host restrictions preventing this. This review frequently cmpares to IAV making the point that IAV infects all these species and IBV doesn’t, but it is not so simple, human adapted IAVs circulate in humans, whilst there is evidence of reverse zoonosis to some mammals, it is limited and not demonstrated for avian species.
- Influenza C virus is not mentioned- yet presumably may share common host restrictions to IBV as a human disease. How about IDV? Should these not be mentioned?
This review would benefit from a review of language and grammar. Particularly checking to remove 'personalisation' of viruses.
Author Response
Pekarek and Weaver Response to Reviewers
Summary- this review attempts to summarize evidence for the host restriction of IBV. This is a unique perspective and it is agreed that more experimental evidence is required to understand this. However, revisions are required and overall, the points made could be clearer. Thank you for your suggestions. As you will see in the manuscript, we removed unnecessary information to better focus on the original question which was what is known about host restriction factors for IBV. The critiques are bolded and our responses and revisions are italicized.
Reviewer 1 Comments
Line 28-30- grammar issues
This was removed to clarify more important introductory information.
Line 31- antigenically characterized? Reference of the split since the 1980s?
This was clarified, now in lines 27-28 of the tracked changes document.
Line 51-3- is it worth noting the 1977 reintroduction of H1N1, ‘psuedo-pandemic’?
This is an interesting idea. For the purposes of this review, we chose to focus on the pandemics that had known non-human origins to introduce the possibility of zoonotic infections crossing the host barrier.
Line 68- ‘accidentally’ this could be elaborated. What is implied?
This was revised to now read as “…not likely to have occurred by random chance alone.” to hopefully clarify this confusion.
Line 105- within or between? Within humans? Between species? Older humans can’t transmit to younger? The point is unclear
This was revised to now read as “…may block transmission into hosts whose upper respiratory tract sialylation patterns contain more α2,3 linked sialic acids.”
Line 132- B-HA compatibility and environment preference- rather than combined, doesn’t B-HA contribute to.
This statement was removed during the revision for a more detailed critical analysis in a new paragraph below.
Can this subject of HA stability and physical environment be expanded. What is known for IAV- more stable Ph/ lower temperature is required for sustained transmission compared to avian IAV. Is this not common with IBV? What about differences in animal environments compared to humans- how would IBV fair in transmission. Is membrane fusion even the main driver of the pH stability? If so, why, are host restriction factors present?
To expand on this, we went into more detail describing the optimal growth characteristics observed for not only IBV, but human and avian origin IAV. However, because this came from a single study comparing the three directly, we were hesitant to draw too many conclusions outside of what was presented. Our conclusion focused on the impact these non-optimal growth conditions likely would have on IBV replication.
Line 136- unusual or belong to a group of viruses that share this replication process?
This was revised to “belong to a group of…”.
Line 140- binds not ‘sit’?
This particular sentence was removed to turn focus to the interaction of the polymerase subunits rather than their positioning on the RNA.
Line 141- description is inaccurate or unclear- the ends of the GENOME rather than RNP, as the polymerase is part of the RNP, combined they make the RNP. The way it is described is unclear.
This was removed during revision.
Line 142- repeats DNA replication machinery point.
Revised to read “B-Pol lacks a critical function needed to express the viral proteins:…” to eliminate any confusion.
Line 151- having given quite a lot of detail about mRNA transcription, why not reduce this detail and provide a short summary of replication also? What is the point trying to be achieved here? That IBV and IAV are similar? Perhaps better to remark if IAV has any host adaptations or not in these processes, compared to IBV?
This was removed to maintain focus on what known host adaptations have been described for nuclear replication of IBV.
Line 174- associated spelling. ANP32 proteins have many activities- ref https://onlinelibrary.wiley.com/doi/10.1002/bies.201400058
This was revised to better address the multiple functions of the proteins while focusing on one we identified as relevant for our review.
Line 185- The major mammalian:avian polymerase restriction as described here is very clear and the understanding is pretty good. However, there may be ADDITIONAL host factors to fine tune the polymerase… A point missed, is that IBV polymerase can function in avian cells. It is avian IAVs that cannot function in mammalian cells. So one may hypothesise that there is no major restriction in reverse zoonosis by the IBV polymerase.
We revised this section with more critical analysis of what is known in this field. To address this, we made sure to explicitly state the avian ANP32 proteins cannot support efficient B-Pol activity. This way, we are not necessarily joining activity and viral replication together. Additionally, we discuss the limitations of not having a better understanding of the effect other possible host factors may have on IBV replication.
Line 196- which protein?
Revised to no longer tease an individual protein, rather focus on “…other mechanisms employed to avoid the human immune system in the absence of these alternative polymerase factors.” We thought this revision would better broadly describe possible host adaptations towards immune evasion not related to B-NS1.
Line 200- TLRs AND RNA helicases
This was removed during revision to focus more on the viral adaptations to evade host immunity rather than describing host detection of viruses.
Line 326- what evidence is there that IBV pol replicates preferentially with human ANP32s?
This was revised as described above in Line 185 to better separate polymerase activity from virus replication.
Figure 2- what about the extracellular environment- Fig 2C touches on HA stability- is this not also relevant to extracellular environment/ transmission?
We revised the figure legend to include the extracellular environment in addition to the intracellular environment to better align with the paragraph of section 2 discussing these environmental factors.
This review reads as a general review of IBV in most places, as opposed to the intention of host restriction. Can this be made clearer throughout the review, with more direct conclusions or hypotheses provided, although this is better provided in the discussion. Perhaps some issue is the level of detail provided to make such a point of host restriction, whilst needed, can it be streamlined to better highlight the point being made?
Thank you for your time in reviewing the manuscript. As you can see, a lot of revision was implemented to ensure the focus of the review (IBV host adaptation) would be clearer and remain the focus of the paper. We also ensured a more critical analysis took place of the existing literature to add further value to the manuscript.
Subjects not addressed by this review which should be considered:
- Can older Influenza B viruses cause pandemics from a lab escape, rather than non-human host?
While this would be something to consider, especially with the possible extinction of the Yamagata lineage, we chose not to include this in the revised manuscript for a similar reason we didn’t include the 1977 H1N1 “pseudo-pandemic.” We wanted to maintain focus on how the host adaptations provide a barrier from non-host spillover, and a lab escape of IBV would not constitute a host jumping event.
- It seems remis to not comment on the possible extinction of the B-Yamagata lineage viruses since 2020. More evidence for lack of animal reservoir?
A paragraph was included in the summary and perspective section discussing the possible extinction of the Yamagata lineage.
- What evidence of reverse zoonosis of IAV is there? Isn’t this also limited? There may be common host restrictions preventing this. This review frequently cmpares to IAV making the point that IAV infects all these species and IBV doesn’t, but it is not so simple, human adapted IAVs circulate in humans, whilst there is evidence of reverse zoonosis to some mammals, it is limited and not demonstrated for avian species.
We added a paragraph discussing evidence of reverse zoonosis that focuses on how host adaptations can help to aid our understanding in the barriers to reverse zoonosis events.
- Influenza C virus is not mentioned- yet presumably may share common host restrictions to IBV as a human disease. How about IDV? Should these not be mentioned?
This was a very insightful thing to consider. We incorporated a new paragraph discussing known host restriction factors against all the other influenza types, including ICV and IDV. While we were unable to find a large amount of literature on either ICV or IDV, this fact alone does add to the potential impact the manuscript could have.
This review would benefit from a review of language and grammar. Particularly checking to remove 'personalisation' of viruses.
To ensure a proper review, we had another person outside of the author list proofread the manuscript for areas that could have been missed by the authors during revision.
Thank you for your helpful revision suggestions. These were suggestions that helped to clarify the focus and maintain the research it throughout the manuscript. Additionally, the thoughtful subjects not addressed helped to enhance the impact of the manuscript to expand on limitations in the IBV research field.

Reviewer 2 Report
Overall, the manuscript provides a comprehensive overview of factors potentially involved in the adaptation of Influenza B viruses (IBV) to replicate in humans. However, there are several shortcomings and mistakes in each section of the manuscript:
Abstract:
The abstract lacks a clear statement of the research objectives and hypotheses, making it challenging for readers to grasp the main focus of the review. The abstract does not mention the limitations of the study, such as the limited research on IBV host determinants.
Introduction:
The introduction could benefit from a more concise and focused presentation of key information. It includes extensive background information that might be better placed in a separate section. The citation format is inconsistent, making it difficult to follow the sources and their relevance to the text.
Hemagglutinin-Specific Adaptations:
The section is informative, but it contains numerous grammatical errors and awkward sentence structures that could be improved for clarity.
While the section discusses receptor binding and hemagglutinin activation, it lacks a critical analysis of the existing research and its limitations.
The information about receptor binding preferences could be organized more clearly to enhance readability.
Nuclear Replication Adaptations:
The section provides detailed information but lacks critical analysis and synthesis of the presented data.
Some sentences are overly complex and might benefit from simplification for better comprehension.
There is a need for more explicit connections between the discussed adaptations and their implications for IBV replication in humans.
Immune Evasion:
This section discusses various aspects of immune evasion by IBV but lacks an integrated analysis of how these strategies collectively contribute to viral replication in humans.
The discussion of immune evasion mechanisms could be more concise and focused on their relevance to host adaptation.
The paragraph about adaptive immunity is somewhat convoluted and might benefit from a clearer structure and organization.
Summary and Perspectives:
The section provides a reasonable summary of the manuscript, but it lacks a clear conclusion that ties together the various factors discussed throughout the review.
There is a need for a more explicit call to action or research recommendations for future studies on IBV host determinants.
The section could be improved with a more critical assessment of the limitations and gaps in the current knowledge regarding IBV adaptations.
General Comments:
The manuscript needs substantial proofreading and editing to correct grammatical errors, improve sentence structure, and ensure clarity.
The organization of the content could be more streamlined to enhance readability and maintain a clear focus on the main research questions and findings.
Throughout the manuscript, there is a lack of direct citations for some specific claims and statements, making it difficult for readers to verify the information presented.
The manuscript could benefit from the inclusion of tables or figures to illustrate key points or data, particularly in the sections discussing viral adaptations.
Overall, the manuscript has potential, but it requires significant revisions to enhance clarity, organization, and critical analysis. Additionally, a more focused research question and clear objectives would strengthen the overall structure of the review.
These are some English mistakes and grammatical errors throughout the manuscript. Proofreading and editing can help enhance the clarity and readability of the text.
Author Response
Pekarek and Weaver Response to Reviewers
Summary- this review attempts to summarize evidence for the host restriction of IBV. This is a unique perspective and it is agreed that more experimental evidence is required to understand this. However, revisions are required and overall, the points made could be clearer. Thank you for your suggestions. As you will see in the manuscript, we removed unnecessary information to better focus on the original question which was what is known about host restriction factors for IBV. The critiques are bolded and our responses and revisions are italicized.
Reviewer 2 Comments
Overall, the manuscript provides a comprehensive overview of factors potentially involved in the adaptation of Influenza B viruses (IBV) to replicate in humans. However, there are several shortcomings and mistakes in each section of the manuscript:
Abstract:
The abstract lacks a clear statement of the research objectives and hypotheses, making it challenging for readers to grasp the main focus of the review. The abstract does not mention the limitations of the study, such as the limited research on IBV host determinants.
To address these revisions, we made the following revisions to this section:
- Added a clear hypothesis statement found in lines 12-13
- Added a clear objective statement in lines 13-14
- Added clear limitations statement found in lines 18-19
Introduction:
The introduction could benefit from a more concise and focused presentation of key information. It includes extensive background information that might be better placed in a separate section. The citation format is inconsistent, making it difficult to follow the sources and their relevance to the text.
To make the introduction section more focused, we removed much of the background information that is not completely relevant to host adaptations of influenza B viruses. We also ensured that our citations would always come at the end of a statement or group of statements coming from our source material rather than leading into the citation with in-text citation phrasing.
Hemagglutinin-Specific Adaptations:
The section is informative, but it contains numerous grammatical errors and awkward sentence structures that could be improved for clarity.
To ensure accuracy and clarity, we had a non-author proofread the revised manuscript to catch mistakes that may have gone unchanged during the revision process.
While the section discusses receptor binding and hemagglutinin activation, it lacks a critical analysis of the existing research and its limitations.
To address this concern, we added a more clear analysis paragraph in the section that can be found in lines 414-788 (tracked changes) of the revised copy. Additionally, we added a table describing how the human respiratory environment is optimal for the various functions of B-HA in the replication cycle.
The information about receptor binding preferences could be organized more clearly to enhance readability.
This paragraph (now lines 363-382) was trimmed of unnecessary background information and structured to focus on the differences of binding preferences of B-HA with that of avian A-HA. Then, we describe the moderate differences in B-HA binding preference between the two lineages.
Nuclear Replication Adaptations:
The section provides detailed information but lacks critical analysis and synthesis of the presented data.
A more detailed analysis paragraph was added from lines 1028-1046 in the revised draft. Additionally, analysis of the lack of connections between measured interactions or activity with general virus replication was added in the two paragraphs above.
Some sentences are overly complex and might benefit from simplification for better comprehension.
During the revision, we made statements more direct and removed some of the passive language that may have been causing confusion.
There is a need for more explicit connections between the discussed adaptations and their implications for IBV replication in humans.
Unfortunately, little has been described connecting the nuclear transport efficiency or B-Pol activity with virus replication directly. We address this in our analysis paragraph from lines 1028-1046
Immune Evasion:
This section discusses various aspects of immune evasion by IBV but lacks an integrated analysis of how these strategies collectively contribute to viral replication in humans.
To improve our analysis, this section underwent many changes. Direct analysis paragraphs were added from lines 1174-1186 discussing why these immune evasion mechanisms signal host adaptation and from lines 1187-1195 discussing why we didn’t consider other known immune evasion mechanisms to have been evaluated as possible host restriction factors.
The discussion of immune evasion mechanisms could be more concise and focused on their relevance to host adaptation.
Much of the mechanism description was cut back or eliminated entirely if it had not previously been evaluated as a possible host restriction factor. We focused more on the interaction of B-NS1 with ISG15 due to its described differences in interaction between different species.
The paragraph about adaptive immunity is somewhat convoluted and might benefit from a clearer structure and organization.
A lot of information was removed from this paragraph, and the information that remained was restructured in a way that the focus remained on the importance of CTL evasion during IBV infection.
Summary and Perspectives:
The section provides a reasonable summary of the manuscript, but it lacks a clear conclusion that ties together the various factors discussed throughout the review.
To address this concern, analysis of the possible host restriction factors at different stages of replication was expanded in each paragraph within the section. The final paragraph (lines 2044-2057) was revised to more clearly connect the importance the interactions at different stages of the virus replication cycle to the possible restricted host relationship between humans and IBV.
There is a need for a more explicit call to action or research recommendations for future studies on IBV host determinants.
Table 2 poses some unanswered research questions in the field that would help to better understand the role these different host factors have on the species determination of IBV.
The section could be improved with a more critical assessment of the limitations and gaps in the current knowledge regarding IBV adaptations.
The paragraph added from lines 1682-1688 highlights the lack of comparison studies investigating the impact between factors from different host species and their role supporting IBV replication. Table 2 also highlights unanswered questions that have not been tested to highlight the current gaps in our understanding.
General Comments:
The manuscript needs substantial proofreading and editing to correct grammatical errors, improve sentence structure, and ensure clarity.
Throughout the revision process, we improved the directness of many of our statements and observations. Additionally, we had a non-author proofread the manuscript to ensure as few mistakes were missed as possible during revision. Further, we cut out information that may have been related to the sections individually but were not relevant to the focus on IBV host adaptations.
The organization of the content could be more streamlined to enhance readability and maintain a clear focus on the main research questions and findings.
Much of the information that did not directly relate to IBV host restriction factors or adaptation were removed to ensure the focus of the manuscript was maintained.
Throughout the manuscript, there is a lack of direct citations for some specific claims and statements, making it difficult for readers to verify the information presented.
To improve this, we went through and removed any in-text citations from the beginning of statements or groups of statements that may have led to confusion of what source material the claims came from. Any claims or statements made from a direct source should now all be cited after the statement or group of statements the material came from.
The manuscript could benefit from the inclusion of tables or figures to illustrate key points or data, particularly in the sections discussing viral adaptations.
Two tables were added to the manuscript. Table 1 was added to highlight the different interactions of B-HA had and their evaluation for host restriction of IBV. Table 2 was added to highlight the areas where questions remain before being able to identify different host factors as species determinants due to a lack of comparison with homologous factors from other possible host species. We considered tables in each of the sections, but didn’t for nuclear factors and immune evasion because we thought there were too many unknowns to be able to identify possible roles as host determinants.
Overall, the manuscript has potential, but it requires significant revisions to enhance clarity, organization, and critical analysis. Additionally, a more focused research question and clear objectives would strengthen the overall structure of the review.
Thank you for your review of our manuscript. The comments and suggestions provided us many actionable items to revise and improve the quality of the manuscript. Specifically, the revised manuscript has become more analytical rather than observational and the clarity of the research focus was enhanced.

Reviewer 3 Report
This manuscript titled “Existing evidence for influenza B virus adaptations to drive replication in humans as the primary host” aims to
attempts to frame the current understanding of IBV adaptations to replication in humans and guide future studies to better understand the unusual human-specific circulation of this influenza virus. The manuscript is well-organized and has certain significance. It was addressed a specific gap in the field, the references are appropriate, However, there are some problems in this manuscript that need to be revised;
1 The language needs considerable attention.
2 Please confirm whether the true ratio of nucleus to other organelles is correct.
Author Response
Pekarek and Weaver Response to Reviewers
Summary- this review attempts to summarize evidence for the host restriction of IBV. This is a unique perspective and it is agreed that more experimental evidence is required to understand this. However, revisions are required and overall, the points made could be clearer. Thank you for your suggestions. As you will see in the manuscript, we removed unnecessary information to better focus on the original question which was what is known about host restriction factors for IBV. The critiques are bolded and our responses and revisions are italicized.
Reviewer 3 Comments
Thank you for your suggestions. As you will see in the manuscript, we removed unnecessary information to better focus on the original question which was what is known about host restriction factors for IBV. The critiques are bolded and our responses and revisions are italicized.
This manuscript titled “Existing evidence for influenza B virus adaptations to drive replication in humans as the primary host” aims to
attempts to frame the current understanding of IBV adaptations to replication in humans and guide future studies to better understand the unusual human-specific circulation of this influenza virus. The manuscript is well-organized and has certain significance. It was addressed a specific gap in the field, the references are appropriate, However, there are some problems in this manuscript that need to be revised;
1 The language needs considerable attention.
Thank you for your suggestion. We went back through and revised the language to be as direct and clear as possible throughout all of the sections.
2 Please confirm whether the true ratio of nucleus to other organelles is correct.
For our diagram, we were trying to visualize some of the possible host restriction factors in the context of the cell. Because of this, some of the scaling of proteins and endosomes to nuclear size is not perfectly scaled for understandability.
Thank you for your suggested revisions for the manuscript. We were able to revise the manuscript in a way that ensured the focus was clear and that the revised manuscript did not veer off-topic as far as the initial draft.

Round 2
Reviewer 2 Report
The manuscript has been improved.